# Randomized Clinical Trials Demonstrate the Safety Assessment of *Alkalihalobacillus clausii* AO1125 for Use as a Probiotic in Humans

**DOI:** 10.3390/microorganisms12112299

**Published:** 2024-11-12

**Authors:** Gissel García, Josanne Soto, Antonio Díaz, Jesús Barreto, Carmen Soto, Ana Beatriz Pérez, Suselys Boffill, Raúl De Jesús Cano

**Affiliations:** 1Pathology Department, Clinical Hospital “Hermanos Ameijeiras”, Calle San Lázaro No 701, Esq. a Belascoaín, Centro Habana, La Habana 10400, Cuba; gisselgarcia2805@gmail.com; 2Clinical Laboratory Department, Clinical Hospital “Hermanos Ameijeiras”, Calle San Lázaro No 701, Esq. a Belascoaín, Centro Habana, La Habana 10400, Cuba; josanne.soto@infomed.sld.cu; 3Statistical Department Clinical Hospital “Hermanos Ameijeiras”, Calle San Lázaro No 701, Esq. a Belascoaín, Centro Habana, La Habana 10400, Cuba; antoniodm@infomed.sld.cu; 4Nutrition Department Clinical Hospital “Hermanos Ameijeiras”, Calle San Lázaro No 701, Esq. a Belascoaín, Centro Habana, La Habana 10400, Cuba; barreto.penie@gmail.com (J.B.); susubc@gmail.com (S.B.); 5Biochemistry Department, Biology Faculty, Havana University Cuba, Calle 25 esquina J Vedado, La Habana 10200, Cuba; carmensoto@fbio.uh.cu; 6Cellular Immunology Laboratory, Virology Department, Tropical Medicine Institute “Pedro Kourí” Autopista Novia del Medio Día Km 6 ½ La Lisa, La Habana 11400, Cuba; anab@ipk.sld.cu; 7Biological Sciences Department, California Polytechnic State University, San Luis Obispo, CA 93407, USA

**Keywords:** *Alkalihalobacillus clausii*, clinical trials, safety, GRAS, probiotic

## Abstract

(1) Background: *Alkalihalobacillus clausii* AO1125 is a Gram-positive, motile, spore-forming bacterium with potential as a probiotic due to its broad-spectrum antimicrobial activity, inhibiting pathogens like *Listeria monocytogenes*, *Staphylococcus aureus*, and *Clostridium difficile*, as well as anti-rotavirus activity. Its resilience in gastrointestinal conditions suggests benefits for gut health. This study evaluates the safety and probiotic potential of *A. clausii* AO1125. (2) Methods: Genome annotation identified genes linked to probiotic traits such as stress resistance, gut colonization, immune modulation, and antimicrobial production. The genome was screened for antibiotic resistance genes using CARD, bacteriocin clusters using BAGEL4, and virulence factors via VFDB. Cytotoxicity was assessed on Vero cells and erythrocytes, and a Phase I, double-blind, placebo-controlled clinical trial was conducted with 99 healthy volunteers (50 AO1125, 49 placebo). (3) Results: Genomic analysis confirmed minimal antibiotic resistance genes and the absence of virulence factors, supporting safety. *A. clausii* AO1125 showed no pathogenicity, cytotoxicity, or hemolytic activity and was well-tolerated in clinical settings, with mild, transient abdominal gas as the most common adverse event. (4) Conclusions: The safety profile and genetic basis for probiotic and antimicrobial properties support *A. clausii* AO1125 as a promising probiotic candidate for gastrointestinal health, warranting further clinical research.

## 1. Introduction

*Alkalihalobacillus clausii* is a Gram-positive, rod-shaped, motile, spore-forming bacterium that was initially classified under the genus *Bacillus* [1], later reassigned to the genus *Shouchella* [2] and is now recognized under the genus *Alkalihalobacillus* based on recent phylogenetic and genomic analyses [3,4,5]. This bacterium is commonly found in soil and is a natural component of the mammalian gastrointestinal microbiome. It is known for its ability to thrive in a wide range of environmental conditions, including both aerobic and anaerobic environments, as well as its preference for alkaline pH, which distinguishes it from many other bacterial species [6,7,8,9].

*A. clausii* is widely recognized for its probiotic potential and has been used in probiotic formulations for over five decades [10,11,12]. It is particularly noted for its resilience to harsh gastrointestinal conditions, such as exposure to gastric acid and bile salts, allowing it to survive and exert its beneficial effects within the host’s intestines [11]. One of the critical features of *A. clausii* is its ability to produce antimicrobial substances, including bacteriocins [13,14], which are effective against a variety of Gram-positive pathogens, such as *Staphylococcus aureus*, *Enterococcus faecium*, and *Clostridium difficile* [14,15,16], rotaviruses [17], as well as preventing gastro-intestinal (GI) infections by competitive adhesion of These antimicrobial properties make *A. clausii* a valuable option for preventing and treating gastrointestinal disorders, particularly those related to antibiotic use [4,9,18].

While there is at least one strain of *A. clausii* (088AE) classified generally recognized as safe (GRAS) for use in humans (GRAS Notice 971), it is important to note that the safety and efficacy of each strain, while widely reported, must be individually assessed [19]. Significant differences exist among strains, which can lead to varying levels of probiotic effectiveness and safety profiles. Therefore, comprehensive evaluations, including phenotypic and genotypic characterizations, as well as clinical trials, are essential to ensure the safety and efficacy of each specific strain before it can be recommended for human use [12,20,21]. This careful assessment is crucial given the increasing use of probiotics in healthcare and the need to ensure that they provide the intended health benefits without adverse effects.

The *A. clausii* strain AO1125 is a newly reported Gram-positive, aerobic, alkalophilic, motile, rod-shaped bacterium from alkaline soils. *A. clausii* AO1125 was isolated from agricultural soil as part of a comprehensive, four-decade effort to identify spore-forming bacteria from diverse ancient and contemporary environments. These varied environments were strategically selected to increase the likelihood of discovering strains with unique characteristics beneficial to human and animal health.

While the probiotic potential of *A. clausii* is widely recognized, little is known about its adhesive properties [22], which play a critical role in its ability to colonize the gastrointestinal tract and compete against pathogens. This study investigates the adhesion potential and cell surface properties of *A. clausii* AO1125. Through a comprehensive genomic analysis, we identified genes associated with adhesins, supporting their potential for epithelial adhesion, extracellular matrix (ECM) binding, and aggregation—properties crucial for effective colonization and probiotic action.

In preliminary studies, we identified genes in *A. clausii* AO1125 linked to probiotic functions, such as stress resistance, adhesion, antimicrobial production, and immune support. These features suggest that AO1125 could be beneficial in managing gut dysbiosis and promoting a balanced microbiome. However, before testing its efficacy as a probiotic in humans, it is necessary to confirm its safety for human use.

This study aims to evaluate AO1125 as a safe strain as a nutritional supplement or food additive. Here we present the results of its phenotypic and genotypic characteristics. To assess safety for human use, we also conducted a Phase I clinical trial in healthy adults. This characterization is essential for understanding the safety and probiotic potential of *A. clausii* AO1125, supporting its future use in human health applications.

## 2. Materials and Methods

### 2.1. Source and Isolation of Strain

Approximately 1 g of agricultural soil from San Luis Obispo, CA, was pulverized mechanically and suspended in 10 mL of 1/2-strength tryptone-glucose-yeast extract (TGY) [23], consisting of 5 g/L tryptone, 3 g/L yeast extract, 1 g/L glucose, and 15 g/L agar at pH 7.5. The suspension was vortexed thoroughly for 1–2 min to disperse any bacterial cells and particles present, then heated at 80 °C for 15 min in a water bath. After heat treatment, the suspension was stored at room temperature for 24 h and then streaked onto TGY agar plates. The plates were incubated aerobically at 28 ± 0.5 °C for 48 h. Following incubation, isolated colonies were observed on the agar plates. Several distinct colonies were selected based on their morphological characteristics. Representative, isolated colonies were sent to MIDI Labs (Newark, DE, USA) for preliminary identification through 16S rRNA gene sequencing, providing an initial taxonomic classification of *A. clausii* based on ribosomal RNA analysis. The strain was deposited in the Agricultural Research Service (ARS) Culture Collection (NRRL) and has been assigned the strain designation NRRL B-68302.

### 2.2. Bacterial Characterization

To identify the bacterial isolate at the species level, we employed next-generation sequencing for whole-genome-based taxonomic identification. The bacterial identification was conducted using the TrueBac ID system (Chun Lab Inc., Seoul, Republic of Korea) [24] and Ortho ANI [25] algorithms.

### 2.3. Culture Conditions of Bacterial Strain

A lyophilized stock culture of strain AO1125 was grown in TGY [23] broth at 28 °C for 24 h and subsequently streaked onto TGY agar to check for purity. Single colonies were used to prepare a seed for large-scale production. The seed culture was diluted 1:10 into a larger volume of TGY and incubated for 3–5 days. The bacterial cell mass was then concentrated by centrifugation at 6000× *g*, lyophilized, and milled into a fine powder. The bulk powder was stored at 120 ± 1 °C in sealed Mylar^®^ bags (polyethylene terephthalate, PET) to prevent exposure to light and moisture. −20 °C and subsequently encapsulated at 2.5 ± 0.1 billion CFU per capsule.

### 2.4. Phenotypic Characterization of the Organism

The putative strain of *A. clausii* AO1125 was characterized based on phenotypic and genotypic methods. Phenotypic methods included macroscopic and microscopic morphology, including the Gram-stain, and enzymatic activity [26].

The enzymatic profile of *A. clausii* AO1025 was determined using the API^®^ ZYM system (BioMérieux, Marcy-l’Étoile, France) following the manufacturer’s protocols. The relative enzymatic activity was determined by comparing the color change in each well with a reference color chart provided in the kit. The color intensity was scored from 0 (no activity) to 5 (maximum activity), and results were recorded. All materials were handled following biosafety guidelines, and appropriate controls were included to validate the results. The results were then compared to a reference database for the identification and metabolic characterization of *A. clausii* AO1025.

*A. clausii* AO1125 was tested for resistance to lysozyme [27] by growing the bacteria overnight in MRS broth and then washing and resuspending them in a PBS solution. To simulate the dilution effect of saliva, bacterial suspensions (10^8^–10^9^ CFU/mL) were placed in 4 mL of a sterile electrolyte solution (SES) with and without 100 mg/L of lysozyme. The survival rate of the bacteria was determined after 30 and 120 min by comparing CFU/mL at these time points to the initial count. All experiments were conducted in triplicate.

To assess tolerance to simulated gastric fluids [28], overnight cultures of the bacteria were harvested, washed, and mixed with pepsin in a 2 × SES solution. The gastric environment was mimicked by progressively lowering the pH from 5.0 to 1.8 using HCl, with the bacteria incubated at each pH level for specific durations. Cell counts were taken at 0, 20, 40, 60, and 90 min to determine survival across time. These experiments were also done in triplicate.

Bile resistance was evaluated by growing the bacteria in broth containing various concentrations of bile (0.06% to 1%). After 24 h of incubation at 37 °C, the bacterial growth was measured by optical density (OD_600_) and compared to a control without bile. Results were expressed as a percentage of growth relative to the control, and all assays were performed in triplicate [29].

Antimicrobial susceptibility testing (AST) was conducted, and results were interpreted in accordance with the guidelines provided by the CLSI M100, 33rd. edition, for the Disk Diffusion Method [30]. The antimicrobials used to assess the AST, including Ampicillin, Chloramphenicol, Clindamycin, Erythromycin, Gentamicin, Kanamycin, Streptomycin, Tetracycline, and Vancomycin

### 2.5. Genotypic Characterization of the Organism

#### 2.5.1. Hybrid Sequencing

Hybrid sequencing was conducted at EzBiome (Gaithersburg, MD, USA). Briefly, genomic DNA was extracted, quantified using the fluorescence-based Qubit dsDNA DNA Quantification System (Thermo Fisher, Waltham, MA, USA), and sequenced using both an Illumina NextSeq 2000 platform (2 × 150 bp paired-end reads) and a Nanopore PromethION R10.4.1 flow cell (Eugene, OR, USA). For Illumina sequencing, libraries were prepared using the NEBNext^®^ Ultra™ II FS DNA Library Prep Kit (Illumina, San diego, CA, USA). Nanopore sequencing libraries were prepared using PCR-free Oxford Nanopore Technologies (Cambridge, MA, USA) Ligation Sequencing Kit (SQK-NBD114.24) with the NEBNext^®^ Companion Module (E7180L) from Oxford Nanopore Technologies (Cambridge, MA, USA), following the manufacturer’s protocols without fragmentation or size selection steps.

#### 2.5.2. Assembly and Annotation

Illumina reads were trimmed to remove adapters using BBDuk from BBtools (BBMap project), followed by quality trimming and filtering with fastp v0.21.1 [31]. Nanopore reads were filtered using Filtlong v0.2.1 (--min_length 1000--keep_percent 95) to remove the 5% lowest-quality reads (Filtlong GitHub). Assembly of Nanopore reads was performed using Flye v2.9.2 [32]. Post-assembly polishing of the Nanopore assembly was done with Pilon v1.23 [33], using the clean Illumina reads under default parameters. Finally, hybrid assembly annotation was carried out using Bakta v1.7.0 [34] and KBase [35].

Assembly quality was evaluated using two complementary methods: UBCG (Up-to-date Bacterial Core Gene) [36] statistics and CheckM [37]. UBCG was used to assess assembly completeness and contamination by quantifying core bacterial genes. UBCG recovery indicates the number of core genes present, reflecting the completeness of the assembly, while UBCG paralog detects the presence of duplicate core genes, indicating potential contamination. Additionally, CheckM was employed to further validate assembly quality by estimating completeness and contamination based on single-copy genes typical of the target genome. High completeness in CheckM suggests the genome is largely intact, while low contamination values indicate minimal extraneous sequences. Together, these methods provide a robust evaluation of assembly quality, ensuring an accurate representation of the target genome with minimal contamination.

### 2.6. Genotypic Identification of the Organism

The genotypic characterization of *A. clausii* AO1125 was performed through whole-genome sequencing and annotation. Genomic DNA from the strain was sequenced and annotated by EzBiome (www.ezbiocloud.net, accessed 5 September 2023) [25,38] using a combination of public [34,39,40,41] and proprietary methods.

The assembled genome of AO1125 was utilized as a query in a BLASTN 2.15.0 [42] search to identify the most closely related strains based on nucleotide sequence similarity. This search yielded a set of aligned sequences from homologous strains, which were then used to construct a phylogenetic tree using the Neighbor Joining model [43]. Strains that clustered within the same branch of the tree as AO1125 were its closest evolutionary relatives, suggesting shared ancestry and similar genomic characteristics. PlasmidFinder was used to detect the presence of plasmids in the assembled genome [44].

The relationship of *A. clausii* AO1125 to other extant probiotic strains was assessed through whole-genome phylogenetic analysis. This was accomplished using the BLAST (Basic Local Alignment Search Tool) platform, which enabled the comparison of the AO1125 genome with a database of known probiotic genomes [42]. By aligning and evaluating sequence similarities, a phylogenetic tree was constructed to elucidate the evolutionary relationships between AO1125 and other probiotic strains. This approach allowed for the identification of the closest genetic relatives and provided insight into the taxonomic positioning of AO1125 within the broader context of probiotic species.

#### 2.6.1. Screening the Annotated Genome for Probiotic-Associated Genes

The complete genome annotation of Alkalihalobacillus clausii AO1125 was analyzed to identify genes linked to beneficial traits that may confer probiotic properties [8,22,45,46,47]. This analysis focused on detecting genes involved in functions such as stress resistance, gut colonization, immune modulation, and antimicrobial compound production, all of which are key to the strain’s potential as a probiotic.

#### 2.6.2. Antimicrobial Resistance Genes

To identify genes with high similarity to previously characterized antibiotic resistance genes, the assembled genome of *A. clausii* AO1125 was analyzed using the Comprehensive Antibiotic Resistance Database (CARD) [48]. This curated database includes acquired antibiotic resistance genes documented in the scientific literature, spanning both Gram-positive and Gram-negative bacteria, including various pathogenic species.

#### 2.6.3. Bacteriocin Gene Clusters

The assembled genome was utilized as input for BAGEL4, a bioinformatics tool designed to identify bacteriocin gene clusters and Ribosomally synthesized and Post-translationally modified Peptides (RiPPs) gene clusters [49]. BAGEL4 analyzes the genomic data to detect potential bacteriocins, which are antimicrobial peptides produced by bacteria, as well as RiPPs, which undergo enzymatic modifications after translation, both of which have significant biotechnological and therapeutic applications [50].

#### 2.6.4. Virulence Factor Genes

The annotated genome of AO1125 was also interrogated for the presence of genes associated with virulence. The assembled genome of A. clausii AO1125 was further evaluated for the presence of known virulence factors using the Virulence Factor Database (VFDB) [51]. A total of 32,670 sequences were searched in this latter database.

#### 2.6.5. Biogenetic Amine-Producing Genes

To investigate the presence of potential biogenic amine-producing proteins, the annotated genome of *A. clausii* AO1125 was systematically examined for genes associated with the production of biogenic amines, which are compounds known to cause adverse reactions when present in high concentrations in food [52]. Specifically, genes encoding enzymes such as decarboxylases (e.g., histidine decarboxylase, tyrosine decarboxylase) were targeted, as these enzymes are involved in the biosynthesis of amines like histamine, tyramine, and putrescine. These compounds have been implicated in foodborne illnesses, allergic reactions, and other adverse health events [53]. Comparative analysis with known biogenic amine-producing bacteria was also performed to evaluate the potential risk associated with the strain.

### 2.7. Hemolytic Activity

Hemolytic activity was measured by hemoglobin release (HR) as described by Lefevre et al. [54]. Briefly, human blood obtained from healthy volunteers was centrifuged (3000× *g* for 5 min) to isolate the erythrocytes. The suspension of fresh erythrocytes obtained was washed and resuspended in 1 mL of buffered phosphate solution (PBS). The concentration of the suspension was adjusted by adding PBS to obtain an absorbance of 0.7 at 490 nm, obtained by adding 1 mL of the standard to 14 mL of distilled water. For the hemolysis test, different concentrations of the compound to be tested were added to the erythrocytes. After 1 h of incubation at room temperature, the samples were centrifuged, and the supernatant was added to the wells of a 96-well polystyrene. The supernatant of erythrocytes in PBS1X was used as a negative control, and the supernatant of erythrocytes in distillate water was employed as a positive control.

For the hemolysis assay, varying concentrations of the test substance were added to a suspension of erythrocytes. After a 1-h incubation at room temperature, the samples were centrifuged, and the supernatant was carefully transferred to a 96-well plate. Phosphate-buffered saline (PBS) was used as a negative control, while distilled water served as a positive control, inducing complete hemolysis. The release of hemoglobin from erythrocytes was measured by recording the absorbance at 490 nm using an automated absorbance microplate reader (BioTek ELx800, Thomas Scientific. Swedesboro, NJ, USA). The percentage of hemolysis was calculated using the following formula:$$HR\%=\left(\frac{\left(ODsample-OD\;negative\;control\right)}{\left(OD\;positive\;control-OD\;negative\;control\right)}\right)\;\times \;100$$

### 2.8. Cytotoxicity in Vero Cells

Viable cells were quantified using the MTT colorimetric assay, which measures the reduction of the tetrazolium dye 3-(4,5-dimethylthiazol-2-yl)-2,5-diphenyltetrazolium bromide (MTT) by mitochondrial enzymes, indicative of metabolic activity [55]. Briefly, 100 µL of Vero cells (10^5^ cells per well) were seeded into 96-well culture plates containing serum-free Dulbecco’s Modified Eagle Medium (DMEM) (Aldrich Millipore-Sigma, St. Louis, MO, USA). The cells were then treated with different concentrations of *A. clausii* AO1125 and a control. After a 30-min incubation at 37 °C and 5% CO_2_, 10 µL of MTT was added to each well, and the cells were incubated for an additional 3 h under the same conditions.

Following incubation, 100 µL of dimethyl sulfoxide (DMSO) was added to dissolve the formazan crystals. Absorbance was measured using an ELISA reader (MRX II, Dynex Technologies, Chantilly, VA, USA) at 550 nm with a 630 nm reference wavelength. The percentage of cellular viability was calculated using the following formula:$$\%\;Viability=\left(\frac{At}{Ac}\right)\;\times \;100$$
where *At* is the absorbance of *A. clausii*-treated cells, and *Ac* is the absorbance of control cells. Control cells were treated with the following excipients from the bacterial capsule: calcium phosphate, magnesium stearate, and colloidal silica.

### 2.9. Human Safety Assessment

A comprehensive human safety assessment of *Alkalihalobacillus clausii* AO1125 was conducted in a randomized, double-blind, placebo-controlled clinical trial to evaluate its probiotic potential in healthy adults. The study protocol was approved by the SF-36.

Ameijeiras Clinical and Surgical Hospital Ethics Committee, under approval number NA241LH0037, dated 15 February 2023, the National Institute of Nutrition of Cuba, and the Cuban Ministry of Health, adhering to Good Clinical Practice protocols and the Declaration of Helsinki [56]. This trial was registered with the Cuban Public Registry of Clinical Trials (RPCEC) under the registration number RPCEC00000429, dated 7 August 2023.

#### 2.9.1. Study Design

The trial involved 100 healthy adult participants, aged 18 years and older, with no distinction of sex or skin color and with baseline hematological, clinical chemistry, and hemodynamic parameters within normal ranges. Participants were randomly assigned to either the *A. clausii* AO1125 group or the placebo group, with 50 individuals allocated to each group. The sample size calculation was based on an expected adverse event rate of 20.0% in the AO1125 group and 0.5% in the placebo group, with a Type I error [57,58] rate of 5% (alpha = 0.05) and a Type II error [58] rate of 20% (80% power). A 10% anticipated dropout rate was also incorporated into the calculation, resulting in a minimum requirement of 41 participants per group. However, it was determined to recruit 50 participants per group, providing a buffer to accommodate any unanticipated withdrawals. Randomization was conducted using the EpiData 3.1 software [59], which ensured unbiased allocation to treatment and placebo groups. Blinding was strictly maintained by dispensing the capsules in identical packaging, preventing both participants and study personnel from distinguishing between the active treatment and the placebo.

The intervention spanned an eight-week period, during which participants in the AO1125 group received daily oral doses of *A. clausii* AO1125. Each capsule contained a standardized dose of 5 × 10^9^ CFUs of the probiotic strain along with excipients, including calcium phosphate, magnesium stearate, and colloidal silica, to ensure the stability and consistency of the formulation. Participants in the placebo group received capsules that contained only the excipients, with no active probiotic, and were identical in appearance to the active treatment capsules to maintain blinding. All participants were instructed to take one capsule each morning, 40 min before breakfast, to standardize timing and optimize absorption in fasting conditions. Adherence was closely monitored through regular checks and participant logs, and any deviations from the dose regimen were documented to ensure compliance.

Before study enrollment, informed consent was obtained from each participant, and eligibility criteria were strictly enforced to include only those with normal hematological, clinical chemistry, and hemodynamic parameters. Exclusion criteria included a history of comorbidities, chronic illnesses, or any deviations from normal health parameters that could potentially interfere with study outcomes or participant safety. Following randomization and baseline screening, the final cohort comprised 50 participants in the AO1125 group and 49 in the placebo group, reflecting one exclusion post-randomization. Baseline characteristics and demographic data of both groups were collected and are summarized in Table 1, demonstrating the comparability of the groups at study entry.

The participant flow, randomization process, and group allocations are visually represented in the CONSORT Flow Diagram (Figure 1) [60].

#### 2.9.2. Safety Assessment

Safety was evaluated through monitoring of participants during the study. This included the documentation and assessment of adverse events, which were recorded in real time and categorized by severity, duration, and potential relationship to the intervention. In addition to adverse event monitoring, changes in hematological, biochemical, and hemodynamic parameters were closely tracked to identify any physiological effects of the intervention. Hematological assessments included a complete blood count (CBC) with differential to monitor for any shifts in white and red blood cell counts, hemoglobin levels, and platelet counts. Biochemical analyses covered key indicators such as liver function tests (ALT, AST), renal function markers (creatinine, BUN), and electrolyte balance, ensuring a thorough overview of metabolic stability. Hemodynamic parameters, including blood pressure and heart rate, were regularly measured to detect any cardiovascular effects.

Overall health status was evaluated using the Health Questionnaire SF-36 (version 2) [35]. This questionnaire was administered at baseline and again at the end of the study, allowing for a comparative analysis of health-related quality of life and any potential impact of *A. clausii* AO1125 on overall wellness.

Clinical evaluations and sample collections for these parameters were conducted at two main time points: baseline (prior to the start of intervention) and at the end of the eight-week study period. These data points provided a comparative basis for assessing any significant changes in health indicators, helping to assess the safety profile of *A. clausii* AO1125 over the course of the study.

### 2.10. Sample Collection, Processing, and Data Management

Sample collection, supplement delivery, and clinical evaluation were conducted at the start of the study (week 1) and at the end (week 8). Blood samples were collected via venipuncture [61], properly identified with the participant’s inclusion number, processed, and aliquoted within one hour for storage and future analyses.

All participant records were maintained in a secure, dedicated database. Access to these records was restricted to study and clinical staff responsible for participant care. The Health and Hospital Administration (HHA) was responsible for managing the security of the information technology infrastructure [62].

### 2.11. Clinical Determinations

Hematology parameters [63] for obtaining standard complete blood count data were assessed through the utilization of a hematologic complex autoanalyzer XN-350 (Roche Diagnostics) [64]. This analysis was performed in strict adherence to the manufacturer’s guidelines, with blood samples collected in K3 EDTA tubes at the baseline (one week before the study commencement) and the study’s conclusion (week 8).

For the evaluation of clinical chemistry parameters, a Cobas 600 modular immunochemical autoanalyzer (Roche Diagnostics) was employed. The analysis was performed on serum samples, following the manufacturer’s recommended protocols [65].

### 2.12. Occurrence of Adverse Events Determination

The occurrence of adverse events [66] was documented by the investigators in the case report forms for each subject adhering to the framework outlined by LeFevre et al. [54]. The relationship between each adverse event and the subject’s involvement in the study was assessed and categorized as improbable, possible, probable, or definite. Additionally, the investigators ranked the severity of each adverse event as mild (with no impediment to daily activities), moderate (resulting in partial limitations to daily activities), or severe (rendering daily activities unattainable) [67].

### 2.13. Statistical Analyses

All clinical data collected were analyzed using Statistical Package for Social Sciences (SPSS) version 23.0. Descriptive statistics were used to characterize the samples. Qualitative variables were summarized in absolute numbers and percentages, while quantitative variables were summarized as mean and standard deviation (SD) for normally distributed data. To compare differences between groups according to qualitative variables, the chi-square test (χ2) or Fisher’s exact test was used, while the Student’s *t*-test was used for age and clinical, hematological, and bioimpedance quantitative variables. The kappa test [68] was applied to assess inter-rater agreement for qualitative variables where applicable.

## 3. Results

### 3.1. Genome Characteristics

The genome of *Alkalihalobacillus clausii* AO1125 is composed of 4,500,056 base pairs (bp), with a size range between 4,197,324 and 4,774,103 bp. The assembly consists of a single contig, indicating a complete or near-complete genome. The GC content is 44.68%, with minor variation from 44.51% to 44.75%. The N50 length matches the total genome size at 4,500,056 bp, which is expected for an assembly with only one contig. UBCG statistics confirm the genome’s quality. UBCG recovery was 100%, with all 92 bacterial core genes present, indicating a complete genome with no missing essential bacterial genes. UBCG paralog detection showed a low level of redundancy, with only 5.43% (5 out of 92 core genes) identified as duplicates, suggesting minimal contamination. Domain analysis verified that 100% of the assembled genome belongs to the bacterial domain, with no detectable contamination from Archaea, Eukarya, or Viruses. These characteristics collectively indicate a high-quality, reliable assembly of the *A. clausii* AO1125 genome, making it suitable for further research and applications.

### 3.2. Species and Strain Identification

Morphological analysis revealed that *A. clausii* AO1125 is a Gram-positive, motile, sporogenous, rod-shaped bacterium. Colonies on TGY agar are typically smooth, white, and opaque. This strain is aerobic and thermotolerant, with an optimal growth temperature range between 25 °C and 45 °C, though it can survive at temperatures as high as 50 °C.

Genotypic analysis using the TrueBacID system (ChunLab Inc., Seoul, Republic of Korea) for species identification revealed that the AO1125 isolate was a strain in the species *A. clausii* at an identity level of 99.99%, with its closest relative being strains 088AE and CSI08 with OrthoANI [25] values of 99.8 and 99.7, respectively.

The enzymatic activity of *A. clausii* AO1125 was assessed using the API ZYM system (Table 2). The strain exhibited strong activity for alkaline phosphatase, esterase (C4), and leucine arylamidase, with an enzymatic score of 5. Moderate activity (score of 3) was observed for lipase (C14), valine arylamidase, and cystine arylamidase. Lower activity (score of 2) was detected for acid phosphatase and naphthol-AS-BI-phosphohydrolase. Notably, *A. clausii* AO1125 showed no activity for enzymes such as trypsin, α-chymotrypsin, or β-glucuronidase.

### 3.3. Putative Probiotic Traits

*Alkalihalobacillus clausii* AO1125 showed tolerance to environmental stressors relevant to the gastrointestinal environment. It achieved 75.3% survival in the presence of lysozyme at a concentration of 25 mg/L after 120 min. Under simulated gastric fluid conditions, the strain exhibited 88.3% survival at pH 4.1 after 20 min, 71.7% at pH 3.0 after 40 min, and 18.6% at pH 2.1 after 60 min. The strain also demonstrated tolerance to bile salts, with 88.3% survival in 0.5% bile salts and 77.4% in 1.0% bile salts after 30 min.

The analysis of the complete genome annotation of *A. clausii* AO1125 identified several genes associated with beneficial traits that may confer probiotic properties. The results, including genes involved in stress resistance, gut colonization, immune modulation, and antimicrobial production, are summarized in Table 3.

### 3.4. Antimicrobial Susceptibility Profile

Antimicrobial susceptibility testing for *A. clausii* AO1125 was conducted using the Kirby–Bauer [69] method. The results are summarized in Table 4.

### 3.5. Antimicrobial Resistance Genes

The results of the interrogation of the CARD database with the genome of *A. clausii* AO1125 yielded a total of 9 identified AMR genes; one in the “Perfect” RGI criterium and eight in the RGI criterium of “Strict” match. The results are summarized in Table 5.

### 3.6. Biogenic Amines: Candidates of Amino Acid Decarboxylases

Genome sequence analysis of *Alkalihalobacillus clausii* AO1125 identified a gene encoding arginine decarboxylase (EC 4.1.1.19), an enzyme involved in the conversion of arginine to agmatine, a precursor in polyamine biosynthesis.

### 3.7. Virulence Factors

The results of the interrogation of the VFDB database with the genome of *A. clausii* AO1125 yielded a total of 12 virulence factors, ranging in “percent identity” from 64.39 to 69.58, with a query coverage between 100% (*gnd* and *htpB* genes) and 52.33% (*cap8D*). The results are summarized in Table 6.

### 3.8. Hemolytic Activity and Cytotoxicity

A two-tailed *t*-test [70] was conducted to compare the hemolysis levels between the *A. clausii* AO1125-treated group and the negative control (untreated cells). The results showed no significant difference in hemolysis levels (*p* > 0.05), supporting that *A. clausii* AO1125 does not cause red blood cell lysis. Similarly, a one-way ANOVA [70] was performed to assess cell Vero cell viability among the *A. clausii* AO1125-treated group, untreated control, and positive control. There was no statistically significant reduction in cell viability in the *A. clausii* AO1125 group compared to the untreated control (*p* > 0.05), indicating that *A. clausii* AO1125 is non-cytotoxic (Figure 2).

### 3.9. Clinical and Hematological Determinations

Clinical and hematological determinations showed normal ranges in all individuals; however, significant differences were detected in some clinical and hematological parameters (Table 7).

To determine the averages of each chemical variable, the Student’s *t*-test for independent samples was employed. On the baseline day, the following variables were statistically significant in both study groups: Total protein (*p* < 0.01), albumin (*p* = 0.008), Cholesterol (*p* = 0.042), Bilirubin D (*p* = 0.01), MHC (*p* = 0.03), and eosinophils (Eo) (*p* = 0.036). Additionally, MPV showed statistical significance only on day 60 (*p* = 0.034). Significant differences between the *Alkalihalobacillus clausii* group and the placebo group between baseline and day 60 were confirmed solely for GGT, both at baseline (*p* = 0.02) and on day 60 (*p* = 0.03).

Student’s *t*-test for paired samples was applied to the *A. clausii* group to evaluate changes between baseline and day 60. The following parameters decreased significantly: Glycemia (*p* < 0.01), Creatinine (*p* < 0.01), and Urea (*p* = 0.016). However, there were significant increases in ALAT (*p* = 0.02) and ASAT (*p* < 0.01).

A similar analysis in the placebo group revealed significant increases in Urea (*p* = 0.012), GGT (*p* = 0.026), Hematocrit (HTC) (*p* = 0.001), and Mean Corpuscular Volume (MCV) (*p* < 0.01). Significant reductions were observed in ALAT (*p* = 0.002), ASAT (*p* < 0.01), Cholesterol (*p* < 0.01), Total Bilirubin (*p* = 0.002), and Direct Bilirubin (*p* = 0.001).

Bioimpedance was measured using weight and Body Mass Index (BMI). The *t*-test for both independent and paired samples did not reveal significant differences between the study groups in these measures (Table 6).

### 3.10. Adverse Effect Determination

The study involved a total of 99 individuals, 49 in the placebo group and 50 in the *A. clausii* group. Most individuals, 83.8% (*n* = 83), did not show any adverse effects (AE) (Table 4). Of these, 74% belonged to the group treated with the probiotic (*n* = 37) and 93.9% to the placebo group (*n* = 46). Significant differences were observed between the treated group and placebo according to the development of AE related to participants in the study (*p* = 0.007) (Table 8).

Among the participants who received the *A. clausii* probiotic capsule, 13 individuals (26%) experienced mild adverse events (AEs). The reported adverse events included gastrointestinal pain or discomfort (*n* = 3), gas (*n* = 10), headache (*n* = 1), constipation (*n* = 3), and diarrhea (*n* = 1) (Table 9).

The study cohort consisted of 60 females and 39 males. Adverse events were observed in 8 females (8/60) and 8 males (8/39). A moderate-intensity adverse event was reported in 1 female from the placebo group (7%), who developed edema, which coincided with a diagnosis of dengue infection. This moderate AE was likely associated with the arbovirus rather than the intervention. None of the participants reported limitations in their daily activities, which is consistent with the mild intensity of the events. The adverse effects diminished over the course of the treatment.

### 3.11. Health Questionnaire Analysis

The survey questions related to the physical and emotional health status of the study subjects were analyzed (SF-36, Questions 1, 6, 7, and 11). In both groups, the behavior of the answers to these questions was very similar. Both the placebo group and the treatment group maintained similar responses at both times (Kappa~1, *p* = 0.000). The significant *p* values represent that the agreement between the responses of the two moments is high in both evaluated groups (Table 10).

## 4. Discussion

Recent studies highlight various strains of *A. clausii* for their roles in boosting immunity and addressing metabolic disorders via microbiota modulation [12,71,72,73] In the present study, we investigated the safety of the probiotic strain *A. clausii* AO1125. This assessment involves thorough evaluation across in vitro and in vivo parameters, ensuring the strain meets safety standards for use as a food or nutritional supplement.

Morphological, cultural techniques, and genotypic characterization confirmed that the strain of *Alkalihalobacillus clausii* AO1125 is a pure culture, and its closest relatives were *Alkalihalobacillus clausii* 088AE and *Bacillus clausii* CSI06. The absence of important adverse effects and the clinical parameters in the normal range of all patients indicate that this probiotic strain could be used in dietary supplements.

### 4.1. Probiotic Characteristics of A. clausii AO1125

*Alkalihalobacillus clausii* exhibits notable probiotic properties [74,75], including the ability to survive in harsh gastrointestinal environments and its non-cytotoxicity, making it safe for human consumption. It shows high resistance to gastric acidity and bile salts, enabling it to pass through the stomach and reach the intestines. Its reported strong adhesion to the intestinal epithelium [22] supports colonization and long-lasting probiotic effects. Several genes in *A. clausii* contribute to these beneficial traits, including those related to antimicrobial production (bacteriocins), stress responses, and immune modulation (Table 3). Enzymes like lysine decarboxylase assist in acid resistance, improving its survival and effectiveness as a probiotic.

*A. clausii* AO1125 demonstrated notable tolerance to various environmental stressors relevant to the gastrointestinal tract, underscoring its potential as a probiotic strain. It exhibited resistance to lysozyme, achieving 75.3% survival at a concentration of 25 mg/L after 120 min.

Under simulated gastric fluid conditions, AO1125 maintained high survival rates at different pH levels. After 20 min at pH 4.1, it showed 88.3% viability. As the acidity increased, survival rates decreased, with 71.7% survival observed after 40 min at pH 3.0 and 18.6% survival after 60 min at pH 2.1.

The strain also displayed tolerance to bile salts, maintaining 88.3% survival after 30 min in 0.5% bile salts and 77.4% survival in 1.0% bile salts. Furthermore, in vitro assays confirmed that *A. clausii* AO1125 is non-cytotoxic to Vero cells and exhibits no hemolytic activity, reinforcing its safety for use as a probiotic.

Based on the findings, *A. clausii* AO1125 demonstrates potential as a probiotic strain due to its ability to withstand key environmental stressors encountered in the gastrointestinal tract [27,76]. Its survival in the presence of lysozyme suggests the ability to survive in the presence of antibacterial enzymes found in human secretions. The strain’s survival under varying pH levels, particularly 88.3% viability at pH 4.1, further highlights its ability to tolerate gastric acidity, although its survival decreases in more acidic conditions (pH 2.1). Additionally, the strain’s tolerance to bile salts, with survival rates of 88.3% and 77.4% in 0.5% and 1.0% concentrations, respectively, indicates its potential for reproducing in the small intestine.

Interrogation of the annotated genome of *Alkalihalobacillus clausii* AO1125 revealed a panel of genes that may provide advantages for both the survival of the bacterium and the health of the host. These genes include those involved in stress resistance, adhesion to intestinal cells, and the production of antimicrobial compounds. Such traits can enhance the ability of AO1125 to persist in the gastrointestinal environment while potentially modulating the host’s immune response and inhibiting pathogenic microbes. The identified genes and their respective gene products are detailed in Table 3.

The absence of cytotoxicity and hemolytic activity in vitro further supports the safety profile of *A. clausii* AO1125, confirming that it is unlikely to cause harm to host cells or induce blood cell damage. Taken together, these results suggest that *A. clausii* AO1125 is a suitable and safe candidate for probiotic applications, with the ability to survive and function effectively within the gastrointestinal environment.

While in vitro simulations of gastric challenges provide valuable insight into the resilience of *Alkalihalobacillus clausii* under conditions mimicking the gastrointestinal tract, we acknowledge that they do not fully capture the complexities of the human microbiome environment. In vitro studies demonstrated the strain’s ability to withstand gastric acid and bile, suggesting its potential to survive passage through the digestive system. These findings alone may not guarantee that a clinically relevant level of viability was maintained within the microbiome by the end of the study. However, previous studies with similar strains have shown consistent viability post-ingestion, which supports our assumption of survival and activity within the microbiome [11,77]. Nonetheless, we agree that direct in vivo measurements of intestinal colonization and viability would provide more robust evidence and are a priority for future research.

The combined results of this study suggest that *A. clausii* AO1125 is a valuable addition to the repertoire of *A. clausii* strains with beneficial properties. Its genetic potential for adaptation to the gut environment, along with its ability to support host health, highlights its suitability as a probiotic strain with applications in improving gastrointestinal health.

### 4.2. Antimicrobial Resistance Genes in A. clausii AO1125

Interrogation of the CARD database with the genome of *A. clausii* AO1025 identified the presence of genetic material corresponding to nine different putative genes that could potentially impart antimicrobial resistance to the strain. Of these, the *Erm* gene was classified as “Perfect” according to the RGI criteria with a 100% match, and the *clbC* gene, which we classified as in the “Strict” criterium with a 97.14% match. All others were classified in the “Strict” criterium with less than 50% homology.

The *Erm* (erythromycin ribosome methylation) gene in *A. clausii* encodes a methyltransferase enzyme that confers resistance to macrolide antibiotics, such as erythromycin, by methylating the 23S rRNA in the 50S ribosomal subunit. This modification reduces the binding affinity of macrolides, lincosamides, and streptogramin B antibiotics, preventing these antibiotics from inhibiting bacterial protein synthesis [78]. In *A. clausii*, the *erm* gene is typically located on the chromosome [79], which is significant because its chromosomal location reduces the likelihood of horizontal gene transfer, a mechanism by which antibiotic resistance can spread to other bacteria [74,78].

The presence of the *erm* gene in *A. clausii* enables the strain to resist macrolide antibiotics, supporting its ability to thrive in environments where such antibiotics are present. This trait is particularly useful in the context of probiotic applications, as it allows *A. clausii* to maintain its beneficial effects on gut flora, even during or after antibiotic treatments. Importantly, *A. clausii* is often used as a probiotic to restore healthy gut microbiota [4], and its intrinsic resistance to macrolide antimicrobials contributes to its resilience in clinical settings.

The *clbC* methyltransferase gene in *A. clausii* AO1125 is most likely chromosomally located, with no plasmids detected during genome analysis. This gene confers resistance to ribosome-targeting antibiotics such as linezolid [80], which can help *A. clausii* survive antibiotic treatment in the host. While this resistance mechanism does not play a major role in typical intestinal survival, it becomes advantageous when antibiotics are present, allowing *A. clausii* to maintain its probiotic effects and colonize the gut during antibiotic therapy. It is noteworthy, however, that this resistance profile was not observed (Table 3), as the Kirby–Bauer assay demonstrated high sensitivity to all classes of antimicrobials tested.

### 4.3. Virulence Genes of A. clausii AO1125

Analysis of the genome annotation database revealed the presence of a gene encoding arginine decarboxylase (EC 4.1.1.19). This gene indicates that the strain can produce biogenic amines, important compounds involved in various metabolic processes. L-arginine serves as a precursor in various metabolic pathways, supporting cell division and growth, and functioning as a source of carbon, nitrogen, and energy. It plays a dual role in modulating immune responses and shaping the gut microbiota, influencing both beneficial microbes and pathogens [81].

Arginine decarboxylase is an enzyme that converts L-arginine into agmatine and carbon dioxide (CO_2_), playing a crucial role in bacterial metabolism and stress response. This reaction is particularly important for helping bacteria survive in acidic environments, such as the stomach or acidic regions of the gastrointestinal tract, because the enzyme consumes protons during the decarboxylation process, thereby helping to neutralize the surrounding acidity. The production of agmatine through this process serves multiple functions, as agmatine acts as a precursor for other polyamines like putrescine, which are involved in cell growth, differentiation, and stabilization of DNA and RNA structures. Additionally, agmatine and its derivative polyamines can support biofilm formation, enhancing the stability and robustness of biofilms and enabling bacteria to adhere to surfaces and persist in challenging environments. Through these mechanisms, arginine decarboxylase not only contributes to pH homeostasis within the cell but also aids in the broader adaptation and survival of bacteria in diverse, often hostile, conditions [82].

### 4.4. Hemolysis and Cytotoxicity of A. clausii AO1125

The absence of hemolytic activity and cytotoxicity in *A. clausii* AO1125, as demonstrated through in vitro assays using Vero cells, represents a critical safety attribute for its use as a food supplement. Hemolytic activity can indicate a potential for damaging red blood cells, which would pose a risk to consumer safety [83,84]. Similarly, the lack of cytotoxic effects is essential to ensure that the strain does not harm mammalian cells, supporting its suitability for human consumption [85,86]. These characteristics are fundamental in the safety assessment of probiotic strains, as they ensure the microorganism can be consumed without adverse effects, thereby fulfilling regulatory and safety standards for use in functional foods and dietary supplements.

Our findings align with those of a previous study that assessed the safety of the *Alkalihalobacillus clausii* UBBC07 strain in a rat model [79]. The absence of hemolytic activity and cytotoxicity in Vero cells confirms that *Alkalihalobacillus clausii* AO1125 is safe. These results are consistent with other clinical trials involving *Bacillus* species, which have demonstrated similar safety profiles [11,17,47,71,87,88,89].

### 4.5. Clinical Study

Overall, the safety of *A. clausii* AO1125 was demonstrated through a thorough evaluation of clinical and hematological parameters. Throughout the study, these variables remained within normal ranges, indicating no adverse effects or deviations from expected physiological norms among participants. The differences observed between the baseline and final study days likely reflect normal physiological variations in metabolism and immune function.

Specifically, slight shifts in parameters such as white blood cell counts, liver enzymes, and other metabolic markers may correspond to typical metabolic changes that occur over time. Additionally, these changes might be influenced by the immune-modulating properties of *Alkalihalobacillus clausii*, which can vary based on the host’s physiological environment. Prior research suggests that probiotics can modulate immune responses by interacting with immune cells or altering the gut microbiota, potentially leading to subtle adjustments in immune parameters [89,90]. These findings are consistent with the general safety profile of *A. clausii* and other *Bacillus* species observed in previous clinical trials.

No serious adverse events (AEs) were reported in this study. Most AEs were mild, affecting only a small percentage (26%) of participants in the treated group. This outcome aligns with findings from other clinical trials involving different strains of *Alkalihalobacillus clausii*. For example, a study assessing the efficacy and safety of *A. clausii* (strains O/C, N/R, SIN, T) in treating acute diarrhea in children in India reported that the treatment was well-tolerated, with an incidence of adverse events (9.7%) like that observed in the placebo group (12.3%) [91]. Similarly, another study evaluating the efficacy and safety of *A. clausii* strains for managing irritable bowel syndrome in children demonstrated that the treatment was well-tolerated, with adverse events observed in both the treated and placebo groups [20].

In our study, all participants in the treated group reported improvements in their well-being after receiving the treatment. Health surveys administered before and after capsule consumption indicated that the overall health status of the treated group remained stable and was not negatively impacted compared to the control group. These observations are consistent with the low incidence of adverse events recorded during the study.

### 4.6. Probiotic Viability Through the Digestive Tract

Maintaining probiotic viability through the gastrointestinal tract is challenging, as many strains encounter harsh conditions such as acidic gastric environments and bile salts. To enhance survival, several formulation strategies have been developed [92,93]. These strategies include drying and lyophilization [94], which help stabilize probiotics by removing moisture and preserving cell integrity during storage and passage through the digestive tract; microencapsulation [95], which protects the probiotic cells within a matrix; and the use of coatings or protective agents that improve tolerance to acidic and bile environments.

For instance, studies have shown that lyophilized probiotics retain higher viability over time compared to fresh cultures [96], as they are better protected from environmental stresses. Microencapsulation in alginate or other biopolymer matrices further enhances survival rates as probiotics pass through the stomach. Additionally, combining probiotics with prebiotic fibers can support their growth once they reach the intestines, thereby enhancing colonization [97]. We have referenced relevant literature on these techniques, as suggested, to provide a comprehensive perspective on methods to improve strain viability. These advancements in formulation represent promising approaches to ensure that probiotics, including *Alkalihalobacillus clausii*, reach the intestines at viable levels.

### 4.7. Limitations of the Study

This study has some limitations that should be noted. One such limitation is the absence of a positive control group, which could have provided a comparative benchmark to better assess the effects of the probiotic candidate. This study, being a Phase I clinical trial, primarily focused on evaluating the safety profile of the probiotic candidate rather than on its efficacy. Consequently, certain design elements—such as the inclusion of a positive control group and a detailed assessment of the probiotic’s impact on participants’ gut microbiome—were not part of this initial investigation. These aspects would be more appropriately addressed in subsequent efficacy-focused trials. Future studies could build on these findings by incorporating such elements to provide a more comprehensive understanding of the probiotic’s potential benefits and mechanisms.

We acknowledge that the intestinal viability of *Alkalihalobacillus clausii* AO1125 was not directly evaluated at the end of the study. However, the safety profile observed is supported by several factors. Firstly, the strain’s resilience was demonstrated in in vitro assays, including tests for bile and gastric acid tolerance, indicating its ability to survive gastrointestinal conditions. Secondly, prior studies with similar strains have shown high viability through the digestive tract, providing further support for its in vivo resilience [8,9,75]. Lastly, the absence of adverse effects was consistent across participants, suggesting that any reduced viability (if present) was not the primary reason for the observed safety profile. These elements collectively support that the lack of adverse events is related to the strain’s safety rather than diminished viability.

Additionally, the epidemiological situation involving outbreaks of dengue or respiratory infections in Cuba could be a limitation of our study. Another limitation was related to the participants’ self-reported benefits during probiotic capsule consumption, such as improved stool regularity with softer stools, relief from lower back pain, and the absence of migraines, with all participants noting an overall improvement after treatment. These data will be considered for inclusion in a questionnaire for future studies.

## 5. Conclusions

Our study establishes the probiotic potential and safety of *Alkalihalobacillus clausii* AO1125, supporting its suitability for use as a dietary supplement. Comprehensive in vitro and in vivo evaluations confirmed its non-pathogenicity, absence of cytotoxicity, and lack of hemolytic activity, fulfilling essential criteria for Generally Recognized As Safe (GRAS) status [95]. In a clinical setting, the strain demonstrated a favorable safety profile, with no serious adverse events and only mild effects reported in a minority of participants.

Given these findings, *A. clausii* AO1125 emerges as a promising candidate for enhancing gastrointestinal health and complementing existing probiotic therapies. Future studies should focus on its potential for immune modulation, metabolic health benefits, and its application in diverse demographic groups. Additionally, exploring synergies with other probiotic strains may unlock broader health applications and enhance its therapeutic value.

## Figures and Tables

**Figure 1 microorganisms-12-02299-f001:**
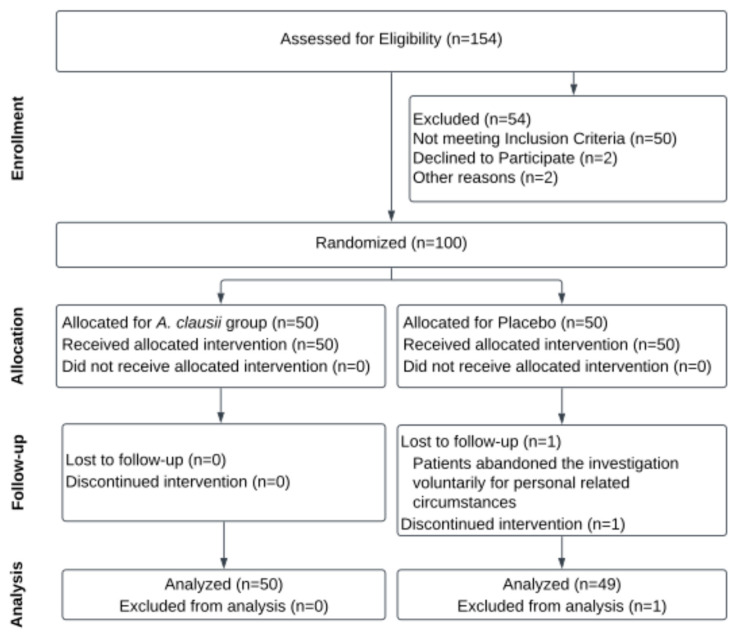
CONSORT Flow Diagram of recruitment and retention throughout the study.

**Figure 2 microorganisms-12-02299-f002:**
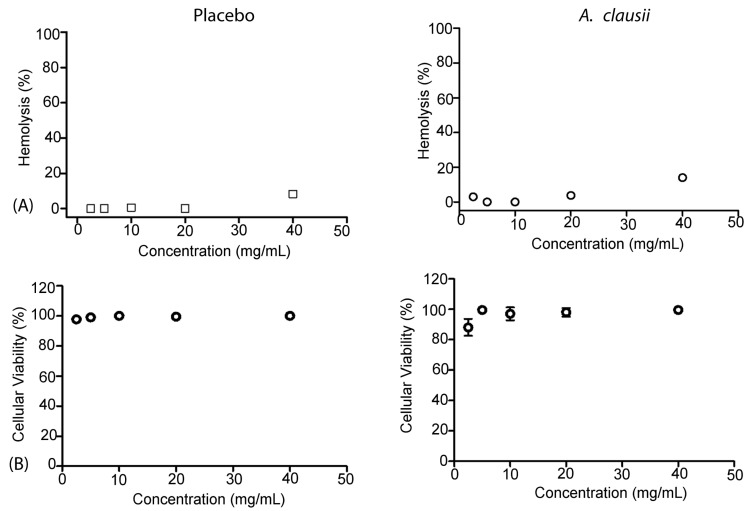
Cytotoxicity of *A. clausii* AO1125. (**A**) Hemolysis induced by *A. clausii* AO1125 on 1% of human erythrocyte solution for 1 h. The hemolytic activity was expressed as % of hemoglobin (hemolysis%). (**B**) Percent of viability induced by *A. clausii* AO1125 in Vero cells. The cytotoxic activity was expressed as % of viability.

**Table 1 microorganisms-12-02299-t001:** Distribution of cases according to demographics parameters.

Demographic Variables		*A. clausii* AO1125 Cohort (*n* = 50)		Placebo Cohort (*n* = 49)		*p* Value
		Number	%	Number	%	
Sex	Female	29	58	31	63.3	0.896 ^a^
	Male	21	42	18	36.7	
Age (years) Median ± SD		50.06 ± 12.87		44.43 ± 14.74		0.185 ^b^
BMI Median ± SD		24.01 ± 3.65		23.63 ± 4.17		0.44 ^b^

^a^ Fisher test; ^b^ *t*-test.

**Table 2 microorganisms-12-02299-t002:** Results of the API^®^ ZYM enzyme profiling to *A. clausii* AO1125.

Enzyme	Result *	Enzyme	Result *
Control	−	Acid phosphatase	+
Alkaline phosphatase	+	Naphthol-AS-BI-phosphohydrolase	+
Esterase (C 4)	+	alpha-Galactosidase	−
Esterase Lipase (C 8)	+	beta-Galactosidase	+
Lipase (C 14)	−	beta-Glucuronidase	−
Leucine arylamidase	−	alpha-Glucosidase	+
Valine arylamidase	−	beta-Glucosidase	+
Cystine arylamidase	−	N-acetyl-beta-glucosaminidase	−
Trypsin	−	alpha-Mannosidase	−
alpha-Chymotrypsin	−	alpha-Fucosidase	−

* '+' indicates enzyme activity; '−' indicates no enzyme activity.

**Table 3 microorganisms-12-02299-t003:** Genes and gene products potentially contributing to probiotic properties in *Alkalihalobacillus clausii* AO1125.

Gene	Encoded Protein
Acid tolerance	
*atpA*	ATP synthase subunit alpha
*atpB*	ATP synthase subunit beta
*ldh1*	L-lactate dehydrogenase 1
*ldhD*	D-lactate dehydrogenase
*pgi*	Glucose-6-phosphate isomerase
*groL*	60 kDa chaperonin
*cspAB*	Cold shock protein
*teaD*	TRAP-T-associated universal stress protein
*ald*	Alanine dehydrogenase
*gabD*	Succinate-semialdehyde dehydrogenase
*fdhD*	Formate dehydrogenase
*pgi*	Glucose-6-phosphate isomerase
*atpF*	ATP synthase subunit B
*arcD1*	Arginine/ornithine antiporter ArcD1
*iPGM*	bisphosphoglycerate-independent phosphoglycerate mutase
*argF*	Ornithine carbamoyltransferase
*gpmI*	2,3-bisphosphoglycerate-independent phosphoglycerate mutase
*argR*	Arginine repressor
Adhesion and aggregation	
*eno*	Enolase
*mntH*	Manganese transferase/Divalent metal cation transporter
*ywgD*	Tyrosine-protein kinase
*tpiA*	Triosephosphate isomerase
*cpsA*	Capsular polysaccharide synthesis protein A
*tuf*	Elongation factor Tu
*spaCBA*	Pilin
Antioxidant defense	
*trxA*	Thioredoxin
*katE*	Catalase
*hemE*	Ferrochelatase
*gsiB*	Glutathione-binding protein GsiB
*sodA*	Superoxide dismutase
Detoxification	
*arsC*	Arsenate reductase
*bshA*	Bile salt hydrolase
*cadA*	Cadmium-transporting ATPase
*hsp18*	18 kDa heat shock protein
Bile tolerance	
*nagB*	Glucosamine-6-phosphate deaminase
*pyrG*	CTP (cytidine triphosphate synthetase) synthase
*nagB*	Glucosamine-6-phosphate deaminase
*bshB*	Bile salt hydrolase
Biofilm formation/Adhesion/Chemotaxis	
*luxS*	Lyase
*cheA*	Chemotaxis protein CheA
*srtA*	Sortase
*hag*	Flagellin
*motB*	Motility protein
*gtfA*	Glycosyltransferase
Carbohydrate metabolism	
*galT*	UDP-glucose--hexose-1-phosphate uridylyl transferase
*nagA*	Glucosamine-6-phosphate deaminase
*bgaB*	Beta-galactosidase
*araC*	Arabinose operon regulatory protein
*xylA*	Xylose isomerase
*galE*	UDP-glucose 4-epimerase
Carbohydrate metabolism	
*dapA*	Dihydropicolinate synthase
*ftsZ*	Cell division protein FtsZ
*ddlA*	D-alanine ligase A
*sigE*	RNA polymerase sigma-E factor
*murA*	DP-N-acetylmuramoyl-L-alanine‚ÄîD-glutamate ligase
Immune modulation	
*hemA*	Glutamyl-tRNA reductase
*groEL*	Chaperonin GroEL
*magl*	Monoacylglycerol lipase
*Hag*	Flagellin
*trpAB*	Tryptophan synthase
Metabolism and Nutrition	
*amyA*	Alpha-amylase
*ldh*	L-lactate dehydrogenase
*lacA*	Beta-galactosidase
*glnA*	Glutamine synthetase
*trpAB*	Tryptophan synthase
*hemL*	Glutamate-1-semialdehyde aminotransferase
*methH*	Methionine synthase
*phoB*	Phosphate regulon protein
*pstS*	Phosphate-binding protein
*lacZ*	Beta-galactosidase
*narG*	Nitrate reductase
*fepA*	Ferrienterobactin receptor
*purA*	Adenylosuccinate synthetase
SCFA (acetate) production	
*ackA*	Acetate kinase
*fabZ*	3-hydroxyacyl-[acyl-carrier-protein] dehydratase
*fabG-1*	3-oxoacyl-[acyl-carrier-protein] reductase
*fabF*	3-oxoacyl-[acyl-carrier-protein] synthase 2
Stress tolerance/Response	
*phoPR*	Two-component response regulator
*treA*	Trehalase
*clpL/yhaX*	Stress response protein
*msrABC*	Methionine sulfoxide reductase
*acoA/yceM*	Oxidoreductase
*yvbW/yhdG*	Amino acid permease
*dnaK*	Chaperone protein DnaK
*GrpE*	Heat shock protein GrpE
*uvrB*	UvrABC system protein B
*msrB*	3-oxoacyl-[acyl-carrier-protein] synthase 2
*fosB*	Metallothiol transferase FosB
*copA*	Copper-exporting P-type ATPase
*emrY*	Multidrug resistance protein
*phoP/Q*	Two-component regulatory system
Vitamin biosynthesis	
*fadD_1*	Long-chain-fatty-acid--CoA ligase
*hpt*	Hypoxanthine-guanine phosphoribosyltransferase
*dfrA*	Dihydrofolate reductase
*thyA2*	Thymidylate synthase
*serA*	D-3-phosphoglycerate dehydrogenase
*dagK*	Diacylglycerol kinase
Bacteriocins	
*gdmA*	Lantibiotic gallidermin
*nisBC*	Nisin biosynthesis protein
*LanKC*	Lanthionine synthetase LanKC
*LanC*	Lantobiotic biosynthesis protein

**Table 4 microorganisms-12-02299-t004:** Results of the antimicrobial susceptibility assay for *A. clausii* AO1125.

	*S. aureus* ATCC 29213		*A. clausii* A01125	
	Diameter (mm) ^1^	Interpretation	Diameter (mm)	Interpretation
Ampicillin	41.8	S ^2^	20.4	R ^4^
Chloramphenicol	24.5	S	32.4	S
Clindamycin	29	S	6.9	R
Erythromycin	28.4	S	6.9	R
Gentamicin	24,2	S	29.1	S
Kanamycin	24	S	28.1	S
Streptomycin	17.1	NA ^3^	20.2	NA
Tetracycline	31.9	S	21.9	S
Vancomycin	18.7	NA	22.9	NA

^1^ Zone diameter, mean value of two or more repeat values., ^2^ S, susceptible. ^3^ NA, Kirby–Bauer disk-based breakpoint not available; and ^4^ R, resistant.

**Table 5 microorganisms-12-02299-t005:** AMR genes detected in the genome of *A. clausii* AO1167B.

RGI Criterium	ARO Term	AMR Gene Family	Drug Class	Resistance Mechanism	% Identity Matching Region
Perfect	*Erm*(34)	Erm 23S ribosomal RNA methyltransferase	Macrolide antibiotic, lincosamide antibiotic, streptogramin antibiotic, streptogramin A antibiotic, streptogramin B antibiotic	Antibiotic target alteration	100
Strict	*vanT* gene in *vanG* cluster	glycopeptide resistance gene cluster, vanT	Glycopeptide antibiotic	Antibiotic target alteration	34.32
Strict	*vanY* gene in *vanM* cluster	vanY, glycopeptide resistance gene cluster	Glycopeptide antibiotic	Antibiotic target alteration	40.67
Strict	*ANT*(4′)-*Ib*	ANT(4′)	Aminoglycoside antibiotic	Antibiotic inactivation	94.14
Strict	*BcIII*	class A Bacillus cereus Bc beta-lactamase	Cephalosporin, penem	Antibiotic inactivation	60.33
Strict	*tetB*(P)	tetracycline-resistant ribosomal protection protein	Tetracycline antibiotic	Antibiotic target protection	38.39
Strict	*vanW* gene in *vanI* cluster	vanW, glycopeptide resistance gene cluster	Glycopeptide antibiotic	Antibiotic target alteration	38.43
Strict	*vanG*	glycopeptide resistance gene cluster, Van ligase	Glycopeptide antibiotic	Antibiotic target alteration	40.17
Strict	*clbC*	Cfr 23S ribosomal RNA methyltransferase	Lincosamide antibiotic, streptogramin antibiotic, streptogramin A antibiotic, oxazolidinone antibiotic, phenicol antibiotic, pleuromutilin antibiotic	Antibiotic target alteration	97.14

**Table 6 microorganisms-12-02299-t006:** Virulence factors detected in the genome of *A. clausii* AO1167B.

VF Factor	VF Gene Name	VF Category	% Identity	Query Cover
VFG048830	*gnd*	Immune Modulation; Antiphagocytosis	66.41	100.0
VFG001855	*htpB*	Adherence; Non-fimbrial adhesin; Cell wall anchored protein	66.35	100.0
VFG002182	*cpsI*	Immune Modulation; Antiphagocytosis	66.13	98.92
VFG000079	*clpC*	Stress survival	69.58	95.36
VFG000077	*clpP*	Stress survival	74.07	95.24
VFG013286	*galE*	Immune Modulation; Inflammatory signaling pathway	63.99	81.91
VFG002158	*lplA1*	Nutritional/Metabolic factor	64.95	76.94
VFG002190	*cpsA*	Immune Modulation;	65.24	75.71
VFG000077	*clpP*	Stress survival	64.39	69.59
VFG001867	*sodB*	Stress survival	65.9	55.01
VFG037100	*msrA/B(pilB*	Stress survival	68.7	54.48
VFG001300	*cap8D*	Immune Modulation	66.91	52.33

**Table 7 microorganisms-12-02299-t007:** Clinical, hematological determinations and bioimpedance parameters.

Clinical Chemistry and Hematology	Normal Range	*A. clausii* Cohort *n* = 50		Placebo Cohort *n* = 49		*p* Value			
		Baseline	Day 60	Baseline	Day 60	p1	p2	p3	p4
Creatinine	47.6–113.4 µmol/L	82.92 ± 16.73	76.35 ± 15.57	73.10 ± 21.59	76.167 ± 17.84	NS	NS	<0.01	NS
Urea	<8.3 mmol/L	4.51 ± 1.34	4.43 ± 1.27	4.02 ± 1.16	4.47 ± 1.12	NS	NS	0.016	0.012
ALAT	<45 U/L	14.28 ± 7.67	16.54 ± 9.37	19.44 ± 9.45	16.26 ± 6.71	NS	NS	0.02	0.002
ASAT	40 U/L	18.33 ± 5.06	21.31 ± 5.91	18.13 ± 4.78	13.35 ± 5.01	NS	NS	<0.01	<0.01
GGT	<50 U/L	15.74 ± 9.02	16.71 ± 8.01	22.57 ± 16.62	24.80 ± 20.21	0.02	0.03	NS	0.026
Total Protein	60–80 g/L	70.54 ± 3.87	70.35 ± 8.54	70.59 ± 6.22	70.77 ± 4.00	<0.01	NS	NS	NS
Albumin	35–52 g/L	44.07 ± 2.51	40.12 ± 5.26	44.78 ± 3.38	44.65 ± 2.89	0.008	NS	NS	NS
Glycemia	4.2–6.1 µmol/L	4.83 ± 0.54	4.43 ± 0.49	4.74 ± 0.55	4.69 ± 0.46	NS	NS	<0.01	NS
Cholesterol	2.81–5.2 mmol/L	4.48 ± 0.95	4.51 ± 0.89	4.95 ± 1.24	4.49 ± 1.11	0.042	NS	NS	<0.01
Triglycerides	0.46–1.8 mmol/L	1.32 ± 0.81	1.22 ± 0.83	1.29 ±0.87	1.33 ± 0.69	NS	NS	NS	NS
Total bilirubin	<17 mmol/L	9.09 ± 4.04	8.82 ± 4.04	9.32 ± 5.33	7.62 ± 4.49	NS	NS	NS	0.002
Direct bilirubin	<5.1 mmol/L	2.89 ± 1.08	2.85 ± 1.44	4.03 ± 4.59	2.95 ± 1.24	0.01	NS	NS	0.001
WBC	(4.5–11) × 10^9^/µL	5.93 ± 1.62	5.85 ± 1.63	6.26 ± 1.62	6.52 ± 1.96	NS	NS	NS	NS
RBC	(F = 4.2–5.4/M = 4.7–6.1) cells/ µL	4.50 ± 0.43	4.59 ± 0.49	4.51 ± 0.40	4.56 ± 0.39	NS	NS	NS	NS
HBG	(F = 12.3–15.3/M = 14.0–17.5) g/dL	135.28 ± 13.60	138.08 ± 15.80	131.53 ± 15.45	132.51 ± 13.71	NS	NS	NS	NS
HTC	(F = 36–45/M = 42–50) %	0.42 ± 0.04	0.42 ± 0.04	0.40 ± 0.04	0.42 ± 0.03	NS	NS	NS	0.001
MVC	80–96.1%	92.46 ± 4.34	90.75 ± 12.13	90.79 ± 6.15	92.30 ± 5.53	NS	NS	NS	<0.01
MCH	33.4–35.5 g/dL	29.5 ± 1.53	29.3 ± 2.91	29.15± 2.36	29.02 ± 2.13	0.03	NS	NS	NS
PLT	(172–450) × 10^3^/mL	233.14 ± 51.6	236.41 ± 62.16	261.88 ± 52.73	259.67± 62.52	NS	NS	NS	NS
RDWCV	(11–14) %	12.78 ± 0.79	12.62 ± 1.92	12.82 ± 1.90	13.12 ± 1.22	NS	NS	NS	NS
MPV	(F: 12–16/M: 14–17.4) g/dL	11.04 ± 1.07	11.20 ± 1.16	10.34 ± 0.90	10.35 ± 0.80	NS	0.034	NS	NS
Neutrophil	1.42–6.34 × 10^9^/L	3.60 ± 1.32	3.61 ± 1.27	3.36 ± 1.16	3.66 ± 1.56	NS	NS	NS	NS
lymphocytes	0.71–4.53 × 10^9^/L	1.64 ± 0.52	1.68 ± 0.48	2.12 ± 0.63	2.10 ± 0.66	NS	NS	NS	NS
Monocytes	0.14–0.72 × 10^9^/L	0.48 ± 0.16	0.50 ± 0.15	0.52 ± 0.13	0.52 ± 0.16	NS	NS	NS	NS
Eosinophils	0–0.54 × 10^9^/L	0.16 ± 0.11	0.15 ± 0.14	0.20 ± 0.17	0.19 ± 0.13	0.036	NS	NS	NS
Basophils	0–0.18 × 10^9^/L	0.02 ± 0.01	0.02 ± 0.01	0.02 ± 0.01	0.02 ± 0.01	NS	NS	NS	NS
Bioimpedance Variables	Baseline	Day 60	Baseline	Day 60		*p* Value			
Weight	68.49 ± 12.09	68.50 ± 12.44	66.64 ± 13.72	66.43 ± 13.57		NS	NS	NS	NS
BMI	24.07 ± 3.65	24.04 ± 3.86	23.63 ± 4.17	2.57 ± 4.15		NS	NS	NS	NS

ALT = alanine transaminase; AST = aspartate transaminase GGT = gammaglutamyl transferase; WBC = White blood cell count; RBC = red blood cell count, HBG = hemoglobin, HTC = hematocrit; MCV = mean corpuscular volume; MCHC = mean corpuscular hemoglobin concentration; PLT = platelet, MPV = mean platelet volume, RDWCV = Red Cell distribution with (Variation in the size of red blood cells); BMI = Body Mass Index F = female, M = male. All values are expressed as mean ± SD.

**Table 8 microorganisms-12-02299-t008:** Association of adverse events with study participation groups.

Presence of AE	*A. clausii* Cohort (*n* = 50)		Placebo Cohort (*n* = 49)		Total	*p* Value
	No	%	No	%	No (%)	
No AE	37	74	46	93.9	83 (83.8)	0.007
AE	13	26	3	6.1	16 (16.16)	
Total	50	100	49	100	99 (100)	

**Table 9 microorganisms-12-02299-t009:** Summary of all adverse effects by intensity.

Adverse Event	*A. clausii* Cohort (*n* = 50)			Placebo Cohort (*n* = 49)		
	Mild	Moderate	Severe	Mild	Moderate	Severe
Abdominal/GI discomfort	3 (6.0)			0	0	0
Acne rosacea	0	0	0	0	0	0
Anxiety Depression	0	0	0	0	0	0
Joint pain	0	0	0	0	0	0
Bronchitis	0	0	0	0	0	0
Excision of birthmarks	0	0	0	0	0	0
Bruises after a fall	0	0	0	0	0	0
Carotid stenosis	0	0	0	0	0	0
Cataract Surgery	0	0	0	0	0	0
Chondrocalcinosis	0	0	0	0	0	0
Colonoscopy and fibroscopy	0	0	0	0	0	0
Cystitis	0	0	0	0	0	0
Dental pain	0	0	0	0	0	0
Diarrhea	1 (2.0)	0	0	2 (2.0)	0	0
Dizziness and nausea	0	0	0	0	0	0
Edema	0	0	0	0	1 (1.0)	0
Gases	10 (20.0)	0	0	0	0	0
General aches	0	0	0	0	0	0
Genital herpes	0	0	0	0	0	0
Headache	1 (2.0)	0	0	0	0	0
Hemorrhoids	0	0	0	0	0	0
Infection	0	0	0	0	0	0
Inflamed prostate	0	0	0	0	0	0
Migraine	0	0	0	0	0	0
Mouth ulcer	0	0	0	0	0	0
Muscle discomfort	0	0	0	0	0	0
Nasal obstruction	0	0	0	0	0	0
Orthopedic pain	0	0	0	0	0	0
Pain following capsule consumption	0	0	0	0	0	0
Palpitations	0	0	0	0	0	0
Radio-infiltration (shoulder)	0	0	0	0	0	0
Rhinitis	0	0	0	0	0	0
Sore throat	0	0	0	0	0	0
Tracheitis	0	0	0	0	0	0
Trouble sleeping (insomnia)	0	0	0	0	0	0
Vaginal dryness	0	0	0	0	0	0
Vagal seizures during or after taking a blood sample.	0	0	0	0	0	0
Vitamin D deficiency	0	0	0	0	0	0
Others (constipation)	3 (6.0)	0	0	0	0	0

GI = gastrointestinal; *n* = number of subjects. Data represent the total number of adverse events followed by % in brackets.

**Table 10 microorganisms-12-02299-t010:** Results and statistical analysis of the Short Form Health Survey (SF-36).

Question	Answer Associated	Placebo (*n* = 49)	Cases (*n* = 50)	Kappa Test/*p* Value	
		Before/After	Before/After	Placebo	Cases
1-General, would you say your health is:	Excellent	5/6	4/6	0.94/0.000	0.73/0.000
	Very Good	20/20	15/18		
	Good	22/21	25/23		
	Fair	2/2	6/3		
6-During the past 4 weeks, to what extent have your physical health or emotional problems interfered with your normal social activities with family, friends, neighbors, or groups?	Not at all	15/ 37	39/41	0.95/0.000	0.81/0.000
	Slightly	14/11	10/9		
	Moderately	18/1	1/0		
	Quite a bit	1/0	0/0		
	Extremely	1/0	0/0		
7-How much bodily pain have you had during the past 4 weeks?	None	14/18	16/16	0.82/0.000	0.78/0.000
	Very mild	14/14	10 /16		
	Mild	19/ 15	17/13		
	Moderate	1/1	6/4		
	Severe	1/1	1/ 1		
11a-I seem to get sick a little easier than other people	Definitely true	0/0	2/1	0.96/0.000	0.86/0.000
	Mostly true	3/3	1/0		
	Don’t know	3/3	4/6		
	Mostly False	10/9	6/6		
	Definitely False	33/34	37/37		
11b-I am as healthy as anybody I know	Definitely true	19/19	13/14	0.97/0.000	0.94/0.000
	Mostly true	16/15	22/ 22		
	Don’t know	7/7	10/10		
	Mostly False	4/4	4/3		
	Definitely False	3/4	1/1		
11c-I expect my health to get worse	Definitely true	1/1	2/2	0.96/0.000	0.94/0.000
	Mostly true	0/0	2/1		
	Don’t know	20/19	14/13		
	Mostly False	5/5	6/7		
	Definitely False	23/24	26/27		
11d-My health is excellent	Definitely true	9/10	12/13	0.93/0.000	0.91/0.000
	Mostly true	27/25	24/23		
	Don’t know	7/7	6/7		
	Mostly False	3/3	4/5		
	Definitely False	3/4	4/2		

## Data Availability

The data supporting the findings of this study are available in the NCBI BioProject database under accession number PRJNA1181829. This BioProject includes the assembled genome, short paired-end reads, and long reads used in this study.

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
