# Peer review of "Randomized Clinical Trials Demonstrate the Safety Assessment of Alkalihalobacillus clausii AO1125 for Use as a Probiotic in Humans"

_microorganisms, 2024, doi:10.3390/microorganisms12112299_

Round 1
Reviewer 1 Report
Comments and Suggestions for Authors
Manuscript review: Randomized clinical trials demonstrate the safety 2 assessment of Alkalihalobacillus clausii AO1125 for use as a 3 probiotic in humans
The manuscript titled above is well-prepared and relevant for publication. However, essential modifications need to be made before it can be published.
ABSTRACT:
- The authors should clarify, in the introductory section of the abstract, the biological potential of the probiotic strain studied. Simply stating that it has antimicrobial properties does not justify conducting a clinical trial with it. Is the antimicrobial action broad or limited? Against which microbial genera is Alkalihalobacillus clausii most effective?
- Was the probiotic’s viability in the gut microbiome evaluated at the end of the study? If so, this is important data that should already be included in the abstract.
- Results related to genetic characterization and antimicrobial resistance should also be explicitly mentioned in the abstract.
INTRODUCTION:
- The authors mention that the probiotic studied is widely used and commonly found in the mammalian microbiome. However, the merit and relevance of the findings are not clear in the justification and objective at the end of the introduction.
- What are the gaps in the literature regarding known and used Alkalihalobacillus clausii strains? What inspired the isolation and study of this new strain? What sets it apart from others, justifying the clinical trial? These questions should be concisely explained in the introduction.
MATERIALS AND METHODS:
- There is a lack of references in this section. Based on which protocols from previous manuscripts did the authors conduct the assays that lack reference?
RESULTS:
- This section is clear and well-detailed.
DISCUSSION:
- The discussion is well-developed but should address the following points:
- What ensures that the absence of clinically relevant adverse effects of Alkalihalobacillus clausii is related to its safety and not reduced viability, given that intestinal viability was not evaluated at the end of the study?
- Do the authors believe that the in vitro evidence, simulating gastric challenge, provides sufficient evidence to believe that the strain maintained a clinically relevant level of viability in the individuals' microbiome by the end of the study? This is an important limitation that should be thoroughly addressed and defended in the discussion.
- Many studies on probiotics report difficulties with the viability of the strains used. The authors should add a reflection on formulation strategies that aim to enhance probiotic viability in the gastrointestinal tract. Refer to the following and other studies on the topic:
- Li C, Wang ZX, Xiao H, Wu FG. Intestinal Delivery of Probiotics: Materials, Strategies, and Applications. Adv Mater. 2024 Aug;36(32)
. doi: 10.1002/adma.202310174. PMID: 38245861.
- Han M, Lei W, Liang J, Li H, Hou M, Gao Z. The single-cell modification strategies for probiotics delivery in inflammatory bowel disease: A review. Carbohydr Polym. 2024 Jan 15;324:121472. doi: 10.1016/j.carbpol.2023.121472. PMID: 37985038.
- Xu C, Guo J, Chang B, Zhang Y, Tan Z, Tian Z, Duan X, Ma J, Jiang Z, Hou J. Design of probiotic delivery systems and their therapeutic effects on targeted tissues. J Control Release. 2024 Nov;375:20-46. doi: 10.1016/j.jconrel.2024.08.037. PMID: 39214316.
CONCLUSION:
- The conclusion is too lengthy. The authors should reduce it to one or two paragraphs, avoiding repetition and discussion of results and instead simply responding to the aims of the study.
- It would be beneficial for the authors to elaborate on more robust future perspectives regarding the use of the strain studied.

Author Response
-
ABSTRACT:
- The authors should clarify, in the introductory section of the abstract, the biological potential of the probiotic strain studied. Simply stating that it has antimicrobial properties does not justify conducting a clinical trial with it. Is the antimicrobial action broad or limited? Against which microbial genera is Alkalihalobacillus clausii most effective?
- We appreciate the input and have revised the Abstract’s (1) Background to read as follows: Alkalihalobacillus clausiiAO1125 is a Gram-positive, motile, spore-forming bacterium with potential as a probiotic due to its broad-spectrum antimicrobial activity, inhibiting pathogens like Listeria monocytogenes, Staphylococcus aureus, and Clostridium difficile as well as anti-rotavirus activity. Its resilience in gastrointestinal conditions suggests benefits for gut health. This study evaluates the safety and probiotic potential of A. clausii AO1125. (lines 26-31)
- Was the probiotic’s viability in the gut microbiome evaluated at the end of the study? If so, this is important data that should already be included in the abstract.
- Unfortunately we did not. We understand this limitation and addressed this shortcoming at the end of the discussion section (lines 742-751)
- Results related to genetic characterization and antimicrobial resistance should also be explicitly mentioned in the abstract.
- Thank you for this observation. We have revised the appropriate section of the Abstract, considering the 200-word limit, as follows: (3) Results: Genomic analysis confirmed minimal antibiotic resistance genes and absence of virulence factors, supporting safety. A. clausii AO1125 showed no pathogenicity, cytotoxicity, or hemolytic activity and was well-tolerated in clinical settings, with mild, transient abdominal gas as the most common adverse event. (lines 36-40)
INTRODUCTION:
- The authors mention that the probiotic studied is widely used and commonly found in the mammalian microbiome. However, the merit and relevance of the findings are not clear in the justification and objective at the end of the introduction.
- What are the gaps in the literature regarding known and used Alkalihalobacillus clausii strains?
- Thank you for pointing out this oversight: We have added the following to the introduction: While the probiotic potential of A. clausii is widely recognized, little is known about its adhesive properties, which play a critical role in its ability to colonize the gastrointestinal tract and compete against pathogens. This study investigates the adhesion potential and cell surface properties of A. clausii AO1125. Through a comprehensive genomic analysis, we identified genes associated with adhesins, supporting its potential for epithelial adhesion, extracellular matrix (ECM) binding, and aggregation—properties crucial for effective colonization and probiotic action. (lines 84-90)
- What inspired the isolation and study of this new strain?
- We now have addressed this comment in: Alkalihalobacillus clausii AO1125 was isolated from agricultural soil as part of a comprehensive, four-decade effort to identify spore-forming bacteria from diverse ancient and contemporary environments. These varied environments were strategically selected to increase the likelihood of discovering strains with unique characteristics beneficial to human and animal health. (lines 78-83)
- What sets it apart from others, justifying the clinical trial? These questions should be concisely explained in the introduction.
- Thank you for your suggestion. We have added text (lines 91-95) addressing this point
MATERIALS AND METHODS:
- There is a lack of references in this section. Based on which protocols from previous manuscripts did the authors conduct the assays that lack reference?
- Thank you for your insightful feedback. We have added pertinent references to the Methods section to enhance the rigor and context of our methodology. References 23-60 now specifically relate to the Materials and Methods section, addressing your concerns and ensuring suitable citations of relevant sources.
RESULTS:
- This section is clear and well-detailed.
DISCUSSION:
- The discussion is well-developed but should address the following points:
- What ensures that the absence of clinically relevant adverse effects of Alkalihalobacillus clausii is related to its safety and not reduced viability, given that intestinal viability was not evaluated at the end of the study?
- Thank you for this observation. It is now addressed in the section 4.7: Limitations of study We acknowledge that intestinal viability of Alkalihalobacillus clausii AO1125 was not directly evaluated at the end of the study. However, the safety profile observed is supported by several factors. Firstly, the strain's resilience was demonstrated in in vitro assays, including tests for bile and gastric acid tolerance, indicating its ability to survive gastrointestinal conditions. Secondly, prior studies with similar strains have shown high viability through the digestive tract, providing further support for its in vivo resilience [8,9,76]. Lastly, the absence of adverse effects was consistent across participants, suggesting that any reduced viability (if present) was not the primary reason for the observed safety profile. These elements collectively support that the lack of adverse events is related to the strain's safety rather than diminished viability. (lines 750 – 759)
- Do the authors believe that the in vitro evidence, simulating gastric challenge, provides sufficient evidence to believe that the strain maintained a clinically relevant level of viability in the individuals' microbiome by the end of the study? This is an important limitation that should be thoroughly addressed and defended in the discussion.
- This is a key point that we failed to address properly. While it is widely reported as a means of assessing probiotic properties of bacteria, these methods can only be used to infer such properties. We hve addressed this issue in section 4.1 of the Discussion. “We have addressed this issue While in vitro simulations of gastric challenges provide valuable insight into the resilience of Alkalihalobacillus clausii under conditions mimicking the gastrointestinal tract, we acknowledge that they do not fully capture the complexities of the human microbiome environment. In vitro studies, demonstrated the strain's ability to withstand gastric acid and bile, suggesting its potential to survive passage through the digestive system. These findings alone may not guarantee that a clinically relevant level of viability was maintained within the microbiome by the end of the study. However, previous studies with similar strains have shown consistent viability post-ingestion, which supports our assumption of survival and activity within the microbiome [11,78]. Nonetheless, we agree that direct in vivo measurements of intestinal colonization and viability would provide more robust evidence and are a priority for future research: (Lines 618-628)
- Many studies on probiotics report difficulties with the viability of the strains used. The authors should add a reflection on formulation strategies that aim to enhance probiotic viability in the gastrointestinal tract. Refer to the following and other studies on the topic:
- Thank you for your observation. We addressed this topic in section 4.6 of the Discussion (Probiotic viability thorough the digestive tract) and found those references that you kindly provided as supporting evidence. (Linen 732-748)
- Li C, Wang ZX, Xiao H, Wu FG. Intestinal Delivery of Probiotics: Materials, Strategies, and Applications. Adv Mater. 2024 Aug;36(32)
. doi: 10.1002/adma.202310174. PMID: 38245861.
- Han M, Lei W, Liang J, Li H, Hou M, Gao Z. The single-cell modification strategies for probiotics delivery in inflammatory bowel disease: A review. Carbohydr Polym. 2024 Jan 15;324:121472. doi: 10.1016/j.carbpol.2023.121472. PMID: 37985038.
- Xu C, Guo J, Chang B, Zhang Y, Tan Z, Tian Z, Duan X, Ma J, Jiang Z, Hou J. Design of probiotic delivery systems and their therapeutic effects on targeted tissues. J Control Release. 2024 Nov;375:20-46. doi: 10.1016/j.jconrel.2024.08.037. PMID: 39214316.
4.7 Probiotic viability through the digestive tract
CONCLUSION:
- The conclusion is too lengthy. The authors should reduce it to one or two paragraphs, avoiding repetition and discussion of results and instead simply responding to the aims of the study.
- It would be beneficial for the authors to elaborate on more robust future perspectives regarding the use of the strain studied.
- We have now reduced the conclusion to two paragraphs pointing out the salient points of the study (Lines 767- 778)
Reviewer 2 Report
Comments and Suggestions for Authors
The manuscript by Gissel García et al. describes the results of the interesting and comprehensive study of Alkalihalobacillus clausii AO1125 probiotic properties.
The strong part of the paper is genomic, in vitro, and clinical investigation of probiotic properties of studied bacteria. However, I must acknowledge that the provided research report has some major issues.
1. First of all is the absence of publically available raw sequencing and genomic data of analyzed bacteria. In the data availability statement, the authors wrote that the data is available from the corresponding author upon a reasonable request. However, the authors do not explain why the data is not publicly available, which does not correspond to the publisher's guidelines and requirements (https://www.mdpi.com/ethics).
Moreover, according to the journal's guidelines manuscripts containing new sequencing data cannot be published until the accession number to the public repository containing the data is provided (https://www.mdpi.com/journal/microorganisms/instructions#sequence).
Thus, I cannot endorse the paper without proper management of data availability.
2. Second, I have not found the citation of the reference to the registration of the described clinical trial study design in an international clinical trials register in the Methods section. This also does not correspond to the journal's instructions (https://www.mdpi.com/journal/microorganisms/instructions).
3. The text of the manuscript needs to be completely revised, as there are some outdated terms used (L. 52 - gastrointestinal microflora), citations of tables that do not occur in the paper (L. 147), repetitions (L. 174-176 and L. 181-183, L. 273-286 and L. 288-296), and misused citations (L. 646).
4. L. 292-293. Please, provide dates and numbers of approval protocols.
5. L. 94. V3-V4 16S rRNA sequencing does not have a proper resolution for taxonomical identification on the species level. How then did the authors manage to identify sequences belonging to Alkalihalobacillus clausii definitively?
The absence of publicly available data on V3-V4 16S rRNA sequences in GenBank or some other data repository (comment 1) makes it difficult for me to find this out by myself.
6. L. 94-95. Please, cite what pairs of primers you used. There are plenty of universal primers for V3-V4 regions.
7. L. 95. Please, describe how PCR products were sequenced.
8. L. 157. Please, provide the proper name and manufacturer of the library preparation kit for nanopore sequencing.
9. L. 158. How the quality of the assembly was assessed? By QUAST, BUSCO, or some other tools? Please, provide this information to the M&M section and properly describe the results of the assembly quality check in the results section. The mention of the contamination level and the quality of the assembly without proper acknowledgment of the criteria used for this is not enough (L. 422-423). By the way, if the quality of the assembly according to BUSCO is 81%, it is debatable whether it is "excellent" or not. The majority of the bioinformaticians would expect 90-95+%.
10. The M&M section contains the description of the results (L. 224-225, L. 282-286).
11. Please, consider joining sections 2.9, 2.10, 2.11, and 2.12. These sections occur to have extensive repetitions (comment 3).
12. Please, consider visualizing the results of your study with figures and avoiding extensive use of tables for these purposes. Genomic (assemble and annotation) data and the results of statistical analysis can be easily visualized with figures.
13. Please, ensure that Figure 2 has a publication-quality resolution (a minimum of 1000 pixels in width/height, or a resolution of 300 dpi or higher [https://www.mdpi.com/journal/microorganisms/instructions]).
14. L. 447-452. Please, support your statements with the results of hypothesis testing.
15. L. 693. The limitation paragraph is very limited. At least, the study design could benefit from the inclusion of a positive control group and the assessment of the studied probiotic candidate on the gut microbiome of participants.
Conclusion. Given the above-described issues of the manuscript, I would recommend the rejection of the manuscript. However, the revision of the paper does not require additional experiments from my point of view, I recommend reconsideration of the paper after major revisions.
Author Response
The manuscript by Gissel García et al. describes the results of the interesting and comprehensive study of Alkalihalobacillus clausii AO1125 probiotic properties.
The strong part of the paper is genomic, in vitro, and clinical investigation of probiotic properties of studied bacteria. However, I must acknowledge that the provided research report has some major issues.
- First of all is the absence of publicly available raw sequencing and genomic data of analyzed bacteria. In the data availability statement, the authors wrote that the data is available from the corresponding author upon a reasonable request. However, the authors do not explain why the data is not publicly available, which does not correspond to the publisher's guidelines and requirements (https://www.mdpi.com/ethics).
We have now revised
Moreover, according to the journal's guidelines manuscripts containing new sequencing data cannot be published until the accession number to the public repository containing the data is provided (https://www.mdpi.com/journal/microorganisms/instructions#sequence).
The sequence data has been submitted to GenBank and are awaiting issuance of Accession Number with release date of November 30, 2025 or upon acceptance of the manuscript, whichever comes first.
Thus, I cannot endorse the paper without proper management of data availability.
- Second, I have not found the citation of the reference to the registration of the described clinical trial study design in international clinical trials register in the Methods section. This also does not correspond to the journal's instructions (https://www.mdpi.com/journal/microorganisms/instructions).
This is presently reported in the front page and now also appears in section 2.9 of the Results section as requested. (Lines 290-297)
- The text of the manuscript needs to be completely revised, as there are some outdated terms used (L. 52 - gastrointestinal microflora)
Than you for pointing out the archaic use of microflora (no excuse). It has now been corrected (see line 54)
citations of tables that do not occur in the paper (L. 147)
These reference to tables in the text have all been addressed. Thank you for pointing them out.
repetitions (L. 174-176 and L. 181-183, L. 273-286 and L. 288-296)
Agreed. We have re-written these sections to reduce redundancy and improve clarity
and misused citations (L. 646).
Reference has been added (Reference [79], (Line 649). Thank you for pointing that out.
- L. 292-293. Please, provide dates and numbers of approval protocols.
Included in the appropriate Materials and Methods section as per request. (Lines 292-295)
- L. 94. V3-V4 16S rRNA sequencing does not have a proper resolution for taxonomical identification on the species level. How then did the authors manage to identify sequences belonging to Alkalihalobacillus clausii definitively?
This information cannot not made available as MIDI Labs did the preliminary identification using their proprietary methods. The text has been changed Representative, isolated colonies were sent to MIDI Labs (Newark, Delaware, USA) for preliminary identification through 16S rRNA gene sequencing, providing an initial taxonomic classification of A. clausiibased on ribosomal RNA analysis (Lines 114)
The absence of publicly available data on V3-V4 16S rRNA sequences in GenBank or some other data repository (comment 1) makes it difficult for me to find this out by myself.
- L. 94-95. Please, cite what pairs of primers you used. There are plenty of universal primers for V3-V4 regions.
We do not have the exact primer sequences used at MIDI. It’s worth noting that the 16S rRNA gene served as one among several methods for the provisional identification of AO1125 and provided guidance for further steps toward definitive identification of the isolate.
- L. 95. Please, describe how PCR products were sequenced.
Similarly, the PCR protocol used at MIDI was not known to us.
- L. 157. Please, provide the proper name and manufacturer of the library preparation kit for nanopore sequencing.
We have now included the following wording: Nanopore sequencing libraries were prepared using the v14 library preparation chemistry (kits SQK-LSK114 and SQK-RBK114) from Oxford Nanopore Technologies (Oxford, United Kingdom), following the manufacturer's protocols without fragmentation or size selection steps. (Lines 172-174)
- L. 158. How the quality of the assembly was assessed? By QUAST, BUSCO, or some other tools? Please, provide this information to the M&M section and properly describe the results of the assembly quality check in the results section. The mention of the contamination level and the quality of the assembly without proper acknowledgment of the criteria used for this is not enough (L. 422-423). By the way, if the quality of the assembly according to BUSCO is 81%, it is debatable whether it is "excellent" or not. The majority of the bioinformaticians would expect 90-95+%.
Point well taken. It was confusing and inaccurate. The text now reads as follows: Assembly quality was evaluated using two complementary methods: UBCG (Up-to-date Bacterial Core Gene) [35] statistics and CheckM [36]. UBCG was used to assess assembly completeness and contamination by quantifying core bacterial genes. UBCG recovery indicates the number of core genes present, reflecting the completeness of the assembly, while UBCG paralog detects the presence of duplicate core genes, indicating potential contamination. Additionally, CheckM was employed to further validate assembly quality by estimating completeness and contamination based on single-copy genes typical of the target genome. High completeness in CheckM suggests the genome is largely intact, while low contamination values indicate minimal extraneous sequences. Together, these methods provide a robust evaluation of assembly quality, ensuring accurate representation of the target genome with minimal contamination (Lines 183-193)
- The M&M section contains the description of the results (L. 224-225, L. 282-286).
Thank you for that observation. The results have been now removed from the M&M section.
- Please, consider joining sections 2.9, 2.10, 2.11, and 2.12. These sections occur to have extensive repetitions (comment 3).
We agree. These sections have been merged as section 2.6.5 (Line 246) and Section 2.92. Safety Assessment (Lines 356-364).
- Please, consider visualizing the results of your study with figures and avoiding extensive use of tables for these purposes. Genomic (assemble and annotation) data and the results of statistical analysis can be easily visualized with figures.
We have given serious consideration your suggestions of providing figures instead of tables and reviewed similar articles for templates and find it difficult to create figures with our tabular data. However, we have created a graphical summary that might address some of your concerns.
- Please, ensure that Figure 2 has a publication-quality resolution (a minimum of 1000 pixels in width/height, or a resolution of 300 dpi or higher [https://www.mdpi.com/journal/microorganisms/instructions]).
It has been verified to be in tiff format, created with Adobe illustrator at 300 dpi.
- L. 447-452. Please, support your statements with the results of hypothesis testing.
Point well taken. We have now revised that section of the results as follows:
A two-tailed t-test was conducted to compare the hemolysis levels between the A. clausii AO1125-treated group and the negative control (untreated cells). The results showed no significant difference in hemolysis levels (p > 0.05), supporting that A. clausii AO1125 does not cause red blood cell lysis. Similarly,
Cytotoxicity (Vero Cell Viability): “A one-way ANOVA was performed to assess cell Vero cell viability among the A. clausii AO1125-treated group, untreated control, and positive control. There was no statistically significant reduction in cell viability in the A. clausii AO1125 group compared to the untreated control (p > 0.05), indicating that A. clausii AO1125 is non-cytotoxic
- L. 693. The limitation paragraph is very limited. At least, the study design could benefit from the inclusion of a positive control group and the assessment of the studied probiotic candidate on the gut microbiome of participants.
We have expanded this section to include lack of a positive control group as well as other limitations. It should be noted, however, this study was designed as a Phase I clinical trial, with a primary emphasis on assessing the safety of the probiotic candidate rather than its efficacy. As such, while the inclusion of a positive control group and an evaluation of the gut microbiome would provide valuable insights, these aspects were beyond the scope of this initial safety-focused trial. Future studies, particularly those focused on efficacy, should incorporate these elements to further explore the probiotic's potential effects on gut health and overall microbiome composition
Conclusion. Given the above-described issues of the manuscript, I would recommend the rejection of the manuscript. However, the revision of the paper does not require additional experiments from my point of view, I recommend reconsideration of the paper after major revisions.
Thank you for your kind comments and thoughtful suggestions. We have tried to address them as best we can.
Round 2
Reviewer 2 Report
Comments and Suggestions for Authors
The authors addressed most of the comments in a good manner.
However, I have not found the accession numbers to publically available sequencing datasets (L. 981-984).
Please, provide accession numbers to raw sequencing data (unassembled raw reads) and assembled genome. As the authors state that they used GenBank for the genomic data repository, it is expected that they should use SRA for raw sequences.
As a reviewer, I cannot endorse the paper without this, as this is a strict requirement from the journal. This is the major issue that needs to be adressed.
Author Response
The authors addressed most of the comments in a good manner.
However, I have not found the accession numbers to publically available sequencing datasets (L. 981-984).
Please, provide accession numbers to raw sequencing data (unassembled raw reads) and assembled genome. As the authors state that they used GenBank for the genomic data repository, it is expected that they should use SRA for raw sequences.
As a reviewer, I cannot endorse the paper without this, as this is a strict requirement from the journal. This is the major issue that needs to be addressed.
Response 1.
The concerns expressed has been addressed in the further revised version of the manuscript in lines 802-804. It now reads: "
Data Availability Statement: The data supporting the findings of this study are available in the NCBI BioProject database under accession number PRJNA1181829. This BioProject includes the assembled genome, short paired-end reads, and long reads used in this study."